# Dietary Interventions for Breast Cancer Prevention: Exploring the Role of Nutrition in Primary and Tertiary Prevention Strategies

**DOI:** 10.3390/healthcare13040407

**Published:** 2025-02-13

**Authors:** Martina Pontillo, Rossella Trio, Nicola Rocco, Ada Cinquerrui, Mariana Di Lorenzo, Giuseppe Catanuto, Francesca Magnoni, Fabrizia Calenda, Carlo Luigi Junior Castiello, Mafalda Ingenito, Alessia Luciana Margherita, Nunzio Velotti, Mario Musella

**Affiliations:** 1Department of Advanced Biomedical Sciences, University of Naples “Federico II”, 80131 Naples, Italy; martina.pontillo@icloud.com (M.P.); fabriziacalenda@gmail.com (F.C.); castiellocarlo2@gmail.com (C.L.J.C.); mafalda.ingenito@outlook.it (M.I.); ale.margherita4@gmail.com (A.L.M.); mario.musella@unina.it (M.M.); 2Breast Unit, University Hospital Federico II, 80131 Naples, Italy; nunzio.velotti@gmail.com; 3Physiology Nutrition Unit, Department of Clinical Medicine and Surgery, University of Naples “Federico II”, 80131 Naples, Italy; rostrio@unina.it (R.T.); mariana.dilorenzo@unina.it (M.D.L.); 4G.Re.T.A. Group for Reconstructive and Therapeutic Advancements, 80123 Naples, Italy; giuseppe.catanuto@humanitascatania.it; 5Department of General Surgery and Medical-Surgical Specialties, University of Catania, 95123 Catania, Italy; ada.cinquerrui@humanitascatania.it; 6Humanitas Istituto Clinico Catanese, Misterbianco, 95045 Catania, Italy; 7Division of Breast Surgery, IEO European Institute of Oncology IRCCS, 20141 Milan, Italy; francesca.magnoni@ieo.it; 8European Cancer Prevention Organization (ECP), 20122 Milan, Italy

**Keywords:** breast cancer, nutrition, diet, prevention, dietary interventions, nutrition and cancer, dietary recommendations for breast cancer, Mediterranean diet, breast cancer risk

## Abstract

**Background:** Breast cancer remains the most prevalent malignancy among women worldwide, necessitating effective prevention strategies. The current literature was scrutinized to investigate the impact of dietary factors, such as the consumption of fruits, vegetables, and whole grains, and dietary patterns such as the Mediterranean diet on reducing the risk of breast cancer. Additionally, the potential role of diet in diminishing the risk of disease recurrence and enhancing outcomes among breast cancer survivors was explored. **Methods**: A systematic literature search was conducted using PubMed, Web of Science, Scopus, and EMBASE to identify relevant studies published between 2000 and 2024. Inclusion criteria were applied to select studies with robust methodologies, including randomized clinical trials, meta-analyses, and prospective cohort studies focusing on adult women. Only studies published in English were considered. Papers on animal studies, editorials, and case series were excluded. **Results**: Our findings show the critical interplay between diet and breast cancer prevention, crucial for the development of effective strategies to both primary and tertiary prevention. Challenges such as adherence to dietary recommendations, cultural and socioeconomic disparities, and limited high-quality evidence were identified. **Conclusions**: This review underscores the critical need to integrate nutrition into clinical practice and highlights avenues for future research, including personalized dietary interventions.

## 1. Introduction

Breast cancer (BC) is the most frequently diagnosed cancer worldwide and remains the leading cause of cancer-related deaths among women [1]. The etiology of BC is multifactorial, encompassing genetic predispositions, behavioral factors, and environmental exposures. Among the modifiable risk factors, diet stands out as one of the most significant in cancer prevention [2].

This review aims to delve into the mechanisms underlying the influence of diet on both the incidence and recurrence of BC. While past investigations primarily focused on examining the impact of specific nutrients or dietary components, recent research has increasingly shifted towards studying dietary patterns. These patterns, which encapsulate a wide range of dietary exposures, offer a more comprehensive understanding of the role of diet in BC.

Indeed, the National Cancer Institute (NCI) highlights that research on the relationship between diet and cancer is increasingly focusing on dietary patterns rather than individual nutrients. This approach provides a more comprehensive understanding of the interactions between different food components and disease risk [3].

Zamora-Ros R. et al. [4] describe that over the past decade, dietary pattern analysis has emerged as a complementary method to better understand cancer risk. The dietary pattern approach considers the entire dietary picture, which is more predictive than analyzing individual foods or nutrients. This addresses the complexity of food consumption and nutrient–food interactions.

By synthesizing extensive dietary data into composite indices or factors, researchers strive to capture the holistic dietary behaviors of individuals. This approach allows for a more nuanced exploration of the relationship between diet and BC risk and recurrence. Despite numerous studies investigating the role of diet in BC prevention and recurrence, there remains a gap in translating these findings into actionable dietary recommendations. This review not only synthesizes the latest evidence on dietary patterns and BC risk but also provides an analysis of both primary and tertiary prevention strategies. By focusing on the interplay between nutrition, genetics, and lifestyle factors, our study highlights the importance of personalized dietary interventions in clinical practice (Figure 1).

## 2. Materials and Methods

According to the PRISMA criteria (Preferred Reporting Items for Systematic reviews and Meta-Analyses), a systematic search was performed of electronic databases (PubMed, Web of Science, Scopus, and EMBASE). We used medical subject headings (MeSH) and free-text words, using the following search terms in all possible combinations: BC, primary prevention, tertiary prevention, diet, and Mediterranean diet. The last search was performed in October 2024. Attention was focused on the following primary outcome, namely, BC prevention, through both primary and tertiary prevention strategies.

From this point of view, only studies reporting on BC incidence and/or prevalence and/or recurrence after appliance of a well-defined prevention strategy were included in this review. The search strategy was limited to articles written in the English language; moreover, papers on animal studies, editorials, and case series were excluded.

N.R. and M.P, two independent authors, analyzed all the papers, selected the suitable manuscripts, and performed the data extraction independently. All duplicate studies were removed. Two other authors (N.V. and M.I.) then checked the eligibility of the studies selected. Discrepancies were resolved by consensus.

The quality of each included study was assessed. For non-randomized studies, the Newcastle–Ottawa Scale (NOS) was used: the NOS contains eight items, categorized into three domains: (1) selection of study (four points); (2) comparability of groups (two points); and (3) ascertainment of exposure and outcomes (three points) for case–control and cohort studies, respectively. 

We identified a total of 1769 articles, of which 106 articles were selected for full-text review. After full-text review, 35 studies were included in the final analysis. The results are summarized in the PRISMA flowchart (Figure 2).

## 3. Results and Discussion

In 2018, the World Cancer Research Fund (WCRF) and the American Institute for Cancer Research (AICR) published “Diet, Nutrition, Physical Activity, and Cancer: A Global Perspective”, known as the WCRF/AICR Third Expert Report [5]. An important point made by the WCRF/AICR Continuous Update Project (CUP) Expert Panel, who authored the Recommendations, was that each recommendation was intended to be one part of a comprehensive package of modifiable lifestyle behaviors that, when taken together, promote a healthy pattern of diet and physical activity conducive to the primary prevention of cancer, other non-communicable diseases, and obesity.

Instead, tertiary prevention can be started following a cancer diagnosis to enhance quality of life and increase the survival rate.

In this context, an adequate assessment of nutritional status, weight management, and the development of a specific nutritional program are essential not only for primary prevention in general, but also for BC survivors to support a healthy lifestyle over the long term [6]. In this regard, the American Society of Clinical Oncology (ASCO) in 2014 promoted an initiative focused on improving access to weight management for cancer survivors [7]. However, a lack of comprehensive information remains in the literature regarding the optimal dietary model for these patients.

Regarding diet, the correlation between dietary consumption and BC might be attributed to the indirect effect of specific nutrients on BC due to their influence on inflammation, DNA damage and repair, oxidative stress, and genetic modifications.

High alcohol consumption is associated with harmful effects on health, some of which include the increased incidence of cancers, liver damage, and cognitive impairment.

Specifically, alcohol consumption is most consistently associated with BC onset [8] and recurrence [9]. In particular, a meta-analysis showed that for every 10 g/day ethanol consumed, there was an increased risk of BC of about 5% and 9% in premenopausal and postmenopausal women, respectively.

Ethanol is a well-known folate antagonist, and the positive association between alcohol and BC might be related to alcohol’s ability to impact on one-carbon metabolism, which is exacerbated by folate deficiency (folate is commonly found in green leafy vegetables). Secondly, Reactive Oxygen Species (ROS) generated during the metabolism of ethanol can induce deleterious epigenetic changes that can promote the incidence of cancers by reducing the expression of tumor suppressor genes. These epigenetic changes include the methylation of gene promoter regions (of tumor suppressor genes) and histone modifications. Although not specific to BC, we know that epigenetic modifications can impact cancer risk, and that certain dietary patterns/foods/nutrients can modify the epigenome [10]. In addition, heavy alcohol consumption has been shown to increase estrogen concentrations through different mechanisms, and, consequently, these steroid hormones may exert a carcinogenic effect on breast tissues [11].

A diet rich in whole grains, fruits, vegetables, and lean proteins could potentially have a protective role for BC risk [12], reducing ROS production and lowering chronic inflammation [13]. Another benefit of the heavy consumption of vegetables and fruits lies in the ability of polyphenols to antagonize the estrogen signaling pathway by either binding estrogen receptors or inhibiting aromatase, which is responsible for estrogen synthesis, thus regulating cancer cell proliferation [14,15]. Likewise, an adequate intake of fiber (at least 10 g/day) may prevent BC risk by reducing estrogen levels and improving insulin sensitivity, thus reducing weight gain, and may also reduce all-cause mortality in BC survivors [16,17,18].

In a prospective study conducted by Ferraro P. et al. [16], a total of 11,576 invasive BC cases in 334,849 EPIC women mostly aged 35–70 y at baseline were identified over a median follow-up of 11.5 y. Dietary fiber was estimated from country-specific dietary questionnaires. BC risk was inversely associated with intakes of total dietary fiber [hazard ratio comparing fifth quintile to first quintile (HR(Q5–Q1)): 0.95; 95% CI: 0.89, 1.01; P-trend = 0.03] and fiber from vegetables (0.90; 0.84, 0.96; P-trend < 0.01).

On the contrary, foods such as refined carbohydrates, saturated fat, red and processed meat, and ultra-processed foods (UPFs) are potentially risky for the onset of BC as they increase the circulating levels of endogenous estrogens, insulin-like growth factor 1 (IGF-1), and pro-inflammatory cytokines [13]. Interestingly, the high-temperature cooking method, rather than red meat itself, is a possible cause of increased BC risk as it promotes the formation of potentially pro-carcinogenic compounds such as poly-aromatic hydrocarbons and heterocyclic amines [19].

Despite the fact that the relationship between BC and soya consumption has become controversial, emerging evidence has shown that the dietary intake of soy foods is associated with a reduction in BC incidence, and women with BRCA mutations might especially benefit from soy intake [20]. A meta-analysis conducted by Magee P.J. et al. [20] examined 14 studies on soy isoflavone intake and its association with BC incidence, involving 369,934 participants and 5828 incident cases of BC. Overall, there was a statistically significant 11% reduction in BC risk among women with high isoflavone intake (summary RR = 0.89; 95% CI: 0.79–0.99) compared with those with the lowest isoflavone intake.

In addition, the aforementioned WCRF/AICR reported that a higher fiber and soy consumption after BC positively improves BC outcomes [21].

Furthermore, ensuring sufficient levels of essential vitamins and minerals either through a balanced diet or supplementation could be beneficial in improving BC outcomes [22] and may also promote overall well-being. For instance, vitamin D deficiency is considered a negative prognostic factor for BC women [23,24], and practice guidelines recommend both vitamin D and calcium supplementation in BC patients to achieve benefit for bone density and minimize fracture risk [25] (Table 1).

In the literature, different dietary patterns have been examined. The Mediterranean diet seems to closely follow these principles. Indeed, adherence to a Mediterranean-style dietary pattern has been suggested as a beneficial factor for preventing BC [24,26].

In a case–control study conducted by Liu, Y. et al. [11], greater adherence to the Mediterranean diet was found to be associated with lower odds of BC. Specifically, a 1-unit increase in the diet score corresponded to a 12% lower likelihood of developing BC.

Despite the fact that some studies suggest that the ketogenic diet (KD) may impact cancer cell metabolism, more research is needed to determine the role of the ketogenic diet in the primary prevention of BC. Recently, the KETOCOMP study evaluated the efficacy of KD based on natural foods in women with BC undergoing curative radiotherapy, highlighting that KD improved body composition with reductions in body weight, mainly due to fat mass loss and the rapid loss of water immediately after starting the diet, in addition to determining favorable hormonal changes [27]. Furthermore, the women experienced significant improvements in emotional and social functioning, sleep quality, and systemic therapy side effects. Similarly, in an RCT conducted by Khodabakhshi, A. et al. [28], women with locally advanced or metastatic BC undergoing planned chemotherapy and receiving a KD showed an improvement in biochemical parameters (i.e., decreased fasting glucose), body composition (i.e., reduced BMI, body weight, and fat mass percentage), and overall survival without substantial side effects on the lipid profile and/or on both liver and kidney damage indicators.

The plant-based diet has been the subject of numerous studies regarding its association with BC prevention. Plant-based diets are low in saturated fats, which are associated with an increased risk of BC. Additionally, they are rich in fiber, which can help to regulate estrogen levels in the body, thus reducing the risk of certain hormone-sensitive breast tumors. In this view, recent evidence suggests that both vegan and plant-based diets with a limitation of both animal products and processed or ultra-processed foods may offer significant benefits in BC prevention and management [29]. Furthermore, epidemiological studies have shown that populations adhering to plant-based diets have lower rates of BC compared to those consuming a typical Western diet. In addition, it has been demonstrated that a whole-food, plant-based diet could lead to beneficial outcomes in women with metastatic BC, particularly in terms of weight management, cardiometabolic health, and hormonal balance [30]. Several mechanisms have been proposed for the beneficial effects of this nutritional approach. Particularly, it has been found that vegan or plant-based diet promotes a healthy gut microbiome, which plays a role in modulating the immune system and reducing inflammation, both critical factors in cancer prevention and management [31]. Further research is needed to fully understand the specific role of the plant-based diet and its interactions with other risk and protective factors (Table 2).

## 4. Challenges and Future Directions

### 4.1. Adherence to Dietary Recommendations

One of the primary challenges in implementing dietary interventions for BC prevention lies in ensuring adherence to recommended dietary patterns.

Many individuals struggle to maintain long-term dietary changes. Factors such as taste preferences, cultural norms, access to healthy foods, and socioeconomic status can influence dietary choices and adherence. Indeed, the association between social position and BC risk is thought to be mediated by well-established risk factors, such as older age at first childbirth, fewer children, and a greater use of hormone replacement therapy (HRT) [32]. Moreover, stress and psychological factors associated with socioeconomic status can also influence eating behaviors, leading to less healthy food choices that are more convenient or comforting.

Addressing these barriers requires multifaceted approaches. This includes education, community-based interventions, and policy changes to promote healthy eating environments.

Warren Andersen S. et al. [33] conducted a prospective cohort study. They investigated adherence to cancer prevention guidelines and its association with cancer risk in low- and middle-income countries. Their findings provided insights into the impact of adherence to dietary and lifestyle recommendations on cancer risk, particularly in regions with varying socioeconomic conditions. The study emphasized the need for targeted interventions that consider the unique challenges faced by different populations.

Shin S. et al. [34] conducted a systematic review and meta-analysis of prospective studies to examine the association between dietary patterns and the risk of BC. By analyzing data from multiple studies, the researchers aimed to identify consistent patterns or correlations between specific dietary habits and the likelihood of developing BC. Concerning the dietary pattern and dietary eating index rather than the consumption of single food groups, they highlighted that a healthy dietary pattern and a healthy eating index were beneficial for reducing BC risk by 38% and even 51%, while an unhealthy dietary pattern could increase BC risk by 44%. These results confirm that lifestyle factors could prevent approximately one-third of BC cases.

In their systematic review and meta-analysis, Shin S. et al. [34] reported that neither Liu et al. [35] nor Aune et al. [36] found a significant association between vegetable consumption and BC risk. This discrepancy may be due to the cooking methods of certain foods; for example, Asians typically consume vegetables after cooking, which might lead to a higher loss of certain nutrients, such as water-soluble and heat-sensitive nutrients, than those available when consuming raw vegetables [37]. Another prospective study conducted by Shin S. et al. [38] highlights that a “Prudent” dietary pattern was not related to lowered BC risk.

Overall, these studies underscore the complexity of dietary adherence in cancer prevention and the necessity of comprehensive, culturally sensitive, and accessible interventions to support individuals in making sustainable dietary changes.

### 4.2. Cultural Considerations and Diversity

Cultural beliefs, traditions, and dietary practices vary widely across populations, influencing food choices and eating habits. Tailoring dietary interventions to accommodate cultural preferences and traditions is essential to maximize their effectiveness and ensure inclusivity. Future research should prioritize understanding the cultural context of dietary behaviors and developing culturally sensitive interventions that resonate with diverse communities. Collaborative efforts involving community stakeholders and cultural leaders can help bridge cultural gaps and promote the acceptance of dietary recommendations.

#### 4.2.1. Traditional Dietary Patterns and BC Risk

Different cultures have distinct dietary patterns that can influence the risk of BCand the effectiveness of dietary interventions. For instance, traditional Asian diets, which are rich in soy products, green tea, and fish, have been associated with lower BC risk. On the contrary, the Western-style diet, typically high in UPFs [39], is associated with an increased risk of elevating cancer risk and particularly BC [40]. In addition, the Western dietary pattern significantly increases mortality risk in BC survivors [41].

Instead, a high-quality diet such as the Mediterranean prototype is linked to reduced BC risk [26] and overall mortality among cancer survivors [41,42], so adapting elements of the Mediterranean diet to local food availability and cultural preferences can be beneficial.

According to Socha et al. [43], postmenopausal women who had undergone mastectomy and adhered to healthy eating habits reported a higher diet-dependent quality of life. Their study highlighted that dietary patterns significantly impacted both the risk of BC recurrence and the overall quality of life. Specifically, diets rich in fruits, vegetables, and whole grains, similar to the Mediterranean diet, were associated with better health outcomes and enhanced quality of life in these women.

#### 4.2.2. Cultural Competence of Healthcare Providers

Healthcare providers must be culturally competent and sensitive to the dietary preferences and restrictions of different cultures. This involves understanding cultural dietary patterns, traditional foods, and cooking methods. Training healthcare providers in cultural competence can improve communication and trust between patients and providers, leading to better dietary adherence. Atomei et al. [44] highlights the importance of integrating cultural dietary habits, traditional food knowledge, and acculturation into dietary counseling to improve outcomes for diverse populations.

#### 4.2.3. Personalized Dietary Recommendations

Providing tailored dietary recommendations that consider cultural foods and cooking methods can help patients adhere to dietary guidelines without feeling deprived. For example, incorporating culturally relevant recipes and cooking techniques can make dietary changes more acceptable and sustainable. Additionally, using visual educational materials, such as videos and cooking manuals that demonstrate how to prepare healthy meals using traditional ingredients, can increase adherence to recommended diets [44].

#### 4.2.4. Community and Cultural Leader Involvement

Engaging cultural leaders and community members in the design and implementation of dietary interventions can enhance their relevance and acceptance. Cultural leaders can act as mediators and health advocates, spreading dietary messages in ways that resonate with the community. For instance, organizing community events that promote healthy and traditional foods, with cooking demonstrations and informative sessions, can facilitate the adoption of healthy eating habits. According to Theodosopoulos et al. [45], especially for migrant populations, who frequently encounter significant obstacles to accessing healthcare, providing culturally sensitive care is not just advantageous but essential. It ensures equity and inclusivity in healthcare delivery, enabling individuals from diverse cultural and linguistic backgrounds to receive care that is both suitable and effective.

### 4.3. Socioeconomic Disparities

Socioeconomic factors such as income, education, and access to healthcare profoundly influence dietary habits and health outcomes, including BC risk. Individuals from disadvantaged socioeconomic backgrounds often face multiple barriers that impede their ability to adopt and maintain healthy dietary behaviors, ultimately exacerbating their risk of BC and other chronic diseases.

For instance, limited financial resources may restrict access to fresh fruits, vegetables, and other nutritious foods, leading to reliance on cheaper, energy-dense, and nutrient-poor options. Food deserts, characterized by a lack of grocery stores and access to healthy food retailers in low-income neighborhoods, further exacerbate disparities in food access and contribute to unhealthy dietary patterns [46].

Moreover, disparities in healthcare access and quality exacerbate existing inequities in BC prevention and management. Individuals from disadvantaged socioeconomic backgrounds may encounter challenges in accessing timely screenings, diagnostic services, and treatment options, leading to delays in cancer detection and poorer outcomes. Structural barriers such as transportation barriers, a lack of health insurance, and the limited availability of culturally and linguistically appropriate healthcare services further contribute to disparities in BC outcomes [47].

To illustrate, let us consider a low-income single mother residing in a rural area with limited access to transportation and healthcare facilities. She may face challenges in affording nutritious foods due to financial constraints and limited availability of grocery stores in her community. Additionally, her ability to attend regular BC screenings and access timely medical care may be hindered by transportation barriers and the lack of health insurance coverage.

Addressing socioeconomic disparities in BC prevention requires a comprehensive approach that tackles structural inequalities and promotes health equity. Policy initiatives aimed at improving food security, such as expanding access to healthy food retail options in underserved communities and implementing nutrition assistance programs, are critical for mitigating socioeconomic disparities in dietary interventions [48].

Empirical evidence indicates that BC rates tend to be higher in wealthier countries, as measured by GDP per capita and the number of Computed Tomography scans performed. These findings are influenced by several socioeconomic factors, primarily concentrated in wealthier countries [49].

Furthermore, efforts to enhance healthcare access and quality, including expanding Medicaid coverage, increasing funding for community health centers, and implementing culturally competent healthcare delivery models, can help ensure equitable access to BC screening, diagnosis, and treatment services [50].

### 4.4. Research Gaps and Methodological Challenges

Despite the growing body of evidence supporting the role of nutrition in BC prevention, several research gaps and methodological challenges persist. Many studies rely on self-reported dietary assessments, which are prone to recall bias and measurement error [51]. For instance, Giovannucci et al. [52] found that fat intake was associated with the risk of BC only when dietary data were retrospectively assessed, highlighting potential biases in study methodologies. Similarly, the Canadian National Breast Screening Study did not demonstrate recall bias [53], suggesting variability in biases across studies. This inconsistency underscores the need for more reliable dietary assessment methods, such as the use of food diaries or technological tools like mobile apps that can provide real-time data collection and reduce recall bias. Moreover, selection bias poses a challenge in case–control studies of diet and cancer, complicating the selection of suitable control groups. Ecologic studies are susceptible to confounding due to population-level correlations, limiting their ability to establish causal relationships. Migrant studies offer insights into environmental factors but may lack specificity regarding the role of diet and cancer [54].

Additionally, the complex interplay between diet, genetics, lifestyle factors, and BC risk necessitates longitudinal studies with large, diverse populations. Future research should prioritize high-quality prospective studies and randomized controlled trials to elucidate causal relationships between dietary factors and BC risk. Advances in biomarker research and omics technologies hold promise for providing insights into the mechanisms underlying dietary influences on BC. Biomarkers such as fatty acid compositions and carotenoid concentrations can serve as valuable indicators in this regard [53].

Additionally, interdisciplinary research combining nutrition science, genetics, and molecular biology can shed light on the biological pathways through which diet influences cancer development. For example, studying gene–diet interactions can help identify individuals who are susceptible to the effects of certain dietary patterns based on their genetic makeup. Collaborative efforts across different fields of study are essential to address the multifaceted nature of diet and cancer risk.

### 4.5. Integration of Multidisciplinary Approaches

BC prevention requires a multidisciplinary approach that incorporates expertise from various fields, including oncology, nutrition science, public health, behavioral psychology, and health communication [55]. Collaborative efforts between researchers, healthcare providers, policymakers, community organizations, and advocacy groups are essential for developing and implementing effective dietary interventions. Interdisciplinary research collaborations can facilitate knowledge exchange, innovation, and the translation of research findings into actionable strategies for BC prevention. Moreover, integrating nutrition education and counseling into routine clinical practice can empower individuals to make informed dietary choices and optimize their breast health [56].

### 4.6. Harnessing Technology and Digital Health Solutions

Technology and digital health solutions offer promising avenues for enhancing the delivery and scalability of dietary interventions for BC prevention. Mobile applications, wearable devices, and online platforms can facilitate self-monitoring, provide personalized dietary recommendations, and support behavior change through interactive tools and social support networks. Leveraging technology-enabled interventions can overcome geographical barriers, reach underserved populations, and empower individuals to take an active role in managing their health. Kirsch E. P. et al. [57] show the importance of incorporating digital health solutions into BC management. The use of digital health platforms and self-monitoring devices improved quality of life and reduced psychological distress. Digital health platforms can also provide important education to patients who may have difficulty accessing care otherwise.

Telemedicine can also facilitate communication between healthcare providers and migrant patients [45]. The providers can offer personalized care by using video consultations. In this way, it is possible to overcome language barriers and provide culturally relevant care, thereby enhancing the overall patient experience [58]. Ensuring the accessibility, usability, and cultural relevance of digital health solutions is paramount to maximizing their impact and promoting health equity.

### 4.7. Ethical and Cultural Considerations in Dietary Recommendations

Promoting specific dietary patterns, particularly when offering personalized recommendations, raises ethical and cultural considerations that warrant careful examination. While evidence suggests the potential benefits of certain diets in reducing BC risk, it is essential to navigate these recommendations with sensitivity to individual preferences, cultural norms, and ethical principles.

One ethical concern is the potential for dietary recommendations to inadvertently perpetuate food-related disparities, particularly among socioeconomically disadvantaged populations [48]. Recommending diets rich in fruits, vegetables, and lean proteins may pose challenges for individuals with limited access to fresh produce or financial constraints. Moreover, cultural dietary practices and traditions vary widely, and imposing Western-centric dietary guidelines without considering cultural context may lead to resistance or alienation within diverse communities.

Furthermore, personalized dietary recommendations based on individual characteristics such as genetic predispositions or metabolic profiles raise ethical questions regarding privacy, autonomy, and informed consent [59]. While personalized nutrition holds promise for tailoring interventions to individual needs, it also introduces complex ethical dilemmas related to data privacy, equity in access to genetic testing, and the potential for unintended consequences, such as stigmatization or discrimination based on genetic information [60].

Culturally sensitive approaches to dietary counseling and intervention are essential to ensure inclusivity and respect for diverse dietary traditions [61]. Healthcare providers and nutrition professionals must engage with patients and communities in culturally competent ways, acknowledging and valuing diverse food practices. Collaborative decision-making processes that prioritize patient autonomy and respect for cultural beliefs can help foster trust and acceptance of dietary recommendations [62].

Moreover, ethical guidelines and professional standards should inform the development and dissemination of dietary recommendations, emphasizing transparency, equity, and patient-centered care [63]. Policies that promote access to nutritious foods, address food insecurity, and support culturally relevant nutrition education initiatives are essential for advancing health equity and reducing disparities in BC prevention.

In conclusion, while dietary recommendations play a crucial role in BC prevention, ethical and cultural considerations must guide their formulation and implementation. By prioritizing inclusivity, cultural sensitivity, and ethical integrity in dietary counseling and intervention, healthcare professionals can ensure that dietary recommendations align with individual preferences, respect cultural diversity, and promote health equity for all populations (Table 3).

## 5. Conclusions

In conclusion, the role of diet in BC prevention emerges as a crucial element in the fight against this widespread disease. The evidence presented clearly demonstrates that a balanced diet rich in fruits, vegetables, whole grains, and lean proteins can help reduce the risk of developing BC, while adopting dietary patterns such as the Mediterranean diet can offer significant benefits. However, challenges related to adherence to dietary recommendations, socioeconomic disparities, and the complexity of interactions between diet, genetics, and lifestyle require a multidisciplinary and personalized approach to BCprevention.

Our work highlights the importance of integrating nutrition into clinical practice and health policy, encouraging interdisciplinary collaborations and joint efforts among researchers, healthcare professionals, policymakers, and communities. Integrating nutrition counseling into standard oncology care and promoting community-based interventions tailored to diverse populations should be considered. Digital health tools, such as mobile apps, can further support patients by facilitating personalized dietary plans and tracking adherence.

Looking to the future, it is essential to pursue high-quality research, involving diverse populations and utilizing innovative approaches such as digital technology to improve the accessibility and effectiveness of dietary interventions. Only through such collaborative efforts can we hope to reverse the rising trend of BC and significantly improve outcomes for women affected by this disease.

## Figures and Tables

**Figure 1 healthcare-13-00407-f001:**
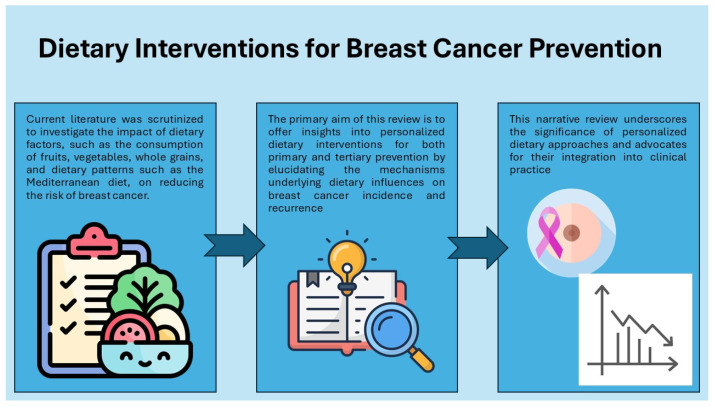
Dietary interventions for BC prevention.

**Figure 2 healthcare-13-00407-f002:**
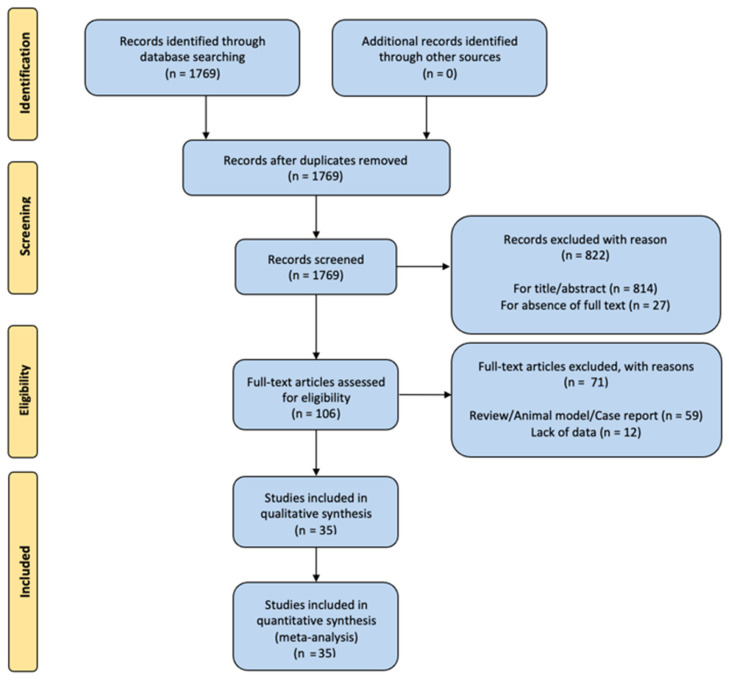
PRISMA flowchart summarizing the study selection process, including the number of articles identified, screened, and excluded, along with reasons for exclusion.

**Table 1 healthcare-13-00407-t001:** Nutrients, mechanisms, and practical implications.

Nutrient	Mechanism of Action	Practical Implications
Fiber	Lowers estrogen levels; improves insulin sensitivity	Recommended ≥10 g/day to reduce BC risk
Polyphenols	Antagonizes estrogen pathways	High fruit/vegetable intake recommended
Vitamin D	Enhances immune regulation	Ensure adequate supplementation for survivors
Soy Isoflavones	Modulates estrogen receptors; inhibits aromatase	Especially beneficial for women with BRCA mutations
Antioxidants	Reduces oxidative stress	Encourage consumption of whole grains, fruits, and vegetables

**Table 2 healthcare-13-00407-t002:** Key dietary patterns and their effects on BC.

Dietary Pattern	Impact on BC Risk	Additional Benefits
Mediterranean diet	Reduces risk by 12% per unit increase	Improves recurrence-free survival; lowers inflammation
Plant-based diet	Lowers risk by regulating estrogen levels	Supports gut health; reduces inflammation
Ketogenic diet	Unclear for prevention; improves body composition during therapy	Enhances emotional and social functioning

**Table 3 healthcare-13-00407-t003:** Barriers to dietary implementation and proposed solutions.

Barrier	Description	Proposed Solutions
Socioeconomic Disparities	Limited access to affordable healthy foods in low-income communities	Expand food subsidies; improve access to fresh produce in underserved areas
Cultural Preferences	Dietary recommendations may conflict with traditional eating habits	Tailor dietary plans to incorporate culturally relevant foods
Adherence Challenges	Difficulty in maintaining long-term dietary changes due to taste preferences	Provide continuous counseling; use mobile apps for tracking and motivation
Educational Gaps	Lack of awareness about the link between diet and breast cancer prevention	Develop community-based educational programs and awareness campaigns
Limited Research on Specific Diets	Unclear evidence for diets like ketogenic diet for all populations	Conduct more randomized trials and cohort studies to build stronger evidence

## Data Availability

Not applicable.

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
