# Peer review of "Dietary Interventions for Breast Cancer Prevention: Exploring the Role of Nutrition in Primary and Tertiary Prevention Strategies"

_healthcare, 2025, doi:10.3390/healthcare13040407_

Round 1
Reviewer 1 Report (Previous Reviewer 2)
Comments and Suggestions for Authors
There are more studies on treatment than prevention of breast cancer today. Therefore, the review is meaningful. Undoubtedly, the role of diet in preventing breast cancer is emerging as an important element in the fight against this common disease. The evidence presented shows that a balanced diet rich in fruits, vegetables, whole grains and lean proteins helps reduce the risk of developing breast cancer. However, studies supporting this situation and demonstrating it statistically should be added. Statistical results that relate nutritional results to breast cancer should be added to the review. Only in this way will the study be more meaningful. It can be re-evaluated after the adjustments are made.
Author Response
Please see the attachment

Reviewer 2 Report (New Reviewer)
Comments and Suggestions for Authors
Review of Manuscript "Dietary Interventions for Breast Cancer Prevention: Exploring the Role of Nutrition in Primary and Tertiary Prevention Strategies"
Overall Evaluation
While the topic of nutrition in breast cancer prevention is highly relevant, the manuscript falls short in several critical areas, including methodological rigor, clarity of presentation, and the novelty of its findings. Below is a detailed assessment of the key weaknesses:
-
Scope and Novelty
- The manuscript does not present novel findings or perspectives. Much of the content reiterates well-established concepts in existing literature, such as the benefits of the Mediterranean diet or the role of plant-based foods. No new insights, frameworks, or hypotheses are introduced, which limits its contribution to the field.
-
Methodology
- The authors state that the review was conducted using databases like PubMed, Scopus, and EMBASE, but there is no detailed description of the search strategy, inclusion/exclusion criteria, or risk of bias assessment. This lack of transparency raises concerns about the reliability of the conclusions drawn.
-
Structure and Clarity
- The manuscript is poorly organized, making it difficult to follow the narrative. The transition between sections is abrupt, and key points are scattered without a cohesive thread tying them together.
- The language is verbose, and many sections repeat concepts unnecessarily (e.g., the benefits of fiber and the Mediterranean diet).
-
Critical Analysis
- While the authors reference numerous studies, there is little critical evaluation of their quality or limitations. For instance, the claims about the ketogenic diet and soy consumption are presented without sufficient scrutiny or acknowledgment of conflicting evidence.
- The manuscript often relies on outdated references, missing recent advancements in the field, such as the impact of gut microbiota on dietary influences in cancer.
-
Practical Implications
- The manuscript emphasizes dietary recommendations but provides little guidance on their implementation in real-world settings. Key challenges, such as cultural differences, socioeconomic barriers, and the feasibility of adherence, are mentioned superficially without actionable solutions.
-
Ethical Considerations
- Ethical concerns, such as the accessibility of recommended dietary patterns for low-income populations, are discussed in a limited capacity. The lack of depth in addressing such critical issues undermines the practical relevance of the manuscript.
Specific Issues
- The abstract does not adequately summarize the findings or significance of the review.
- References are inconsistent in their formatting and do not include recent landmark studies.
- Tables or figures summarizing the dietary interventions or study findings would enhance clarity but are missing.
Recommendation
Due to the aforementioned weaknesses, I recommend rejecting this manuscript in its current form. Significant revisions and additional research are needed to enhance its methodological robustness, provide critical insights, and ensure practical applicability.
Author Response
Please see the attachment

Reviewer 3 Report (New Reviewer)
Comments and Suggestions for Authors
The narrative review correctly describes the issue of breast cancer in women and the relationship between diet and prevention. This narrative review underscores the significance of personalized dietary approaches and advocates for their integration into clinical practice. As the authors emphasize, this review aims to delve into the mechanisms underlying the influence of diet on both the incidence and recurrence of breast cancer.
The subject of the review addresses an important issue in the field of public health, which is why the article is worth publishing. The abstract encourages you to read the full text of the article. The main text describes the problem addressed by the authors in a comprehensive and logical manner. Conclusions of the narrative review described correctly.
Minor comments and tips that will improve the article:
Line 40: Too many keywords, guidelines require up to ten pertinent keywords, without quotation marks
Tables would increase the quality of the article. You can add a table showing selected articles for the narrative review and the results of the studies performed, as well as a table showing the impact of diet on reducing the risk of breast cancer.
There is no Materials and Methods subsection. There is no description of how and where the articles used in the narrative review were searched. This information is included in the abstract, but it is not in the main text of the article. It is not known what the inclusion and exclusion criteria for the articles were, etc. How many articles were found and how many were described. The narrative review methodology needs to be completed.
Round 2
Reviewer 1 Report (Previous Reviewer 2)
Comments and Suggestions for Authors
The recent changes made to the article have made it more meaningful. However, there is no need to give subheadings in the discussion section. Subheadings disrupt the integrity of the discussion and cause too much repetition. The discussion section should be reorganized. After this correction, the article can be published.
Author Response
The recent changes made to the article have made it more meaningful. However, there is no need to give subheadings in the discussion section. Subheadings disrupt the integrity of the discussion and cause too much repetition. The discussion section should be reorganized. After this correction, the article can be published.
We sincerely appreciate the reviewer’s positive feedback and constructive suggestions. We removed the subheadings in the discussion section and reorganized the content to ensure a smoother and more coherent flow while avoiding unnecessary repetition.
Reviewer 2 Report (New Reviewer)
Comments and Suggestions for Authors
Substantially revised, but need minor revision:
Add the statements of novelty at the end of introduction.
Please, Add 2 figure mechanism of the topic that the manuscript discuss for.
Author Response
Substantially revised, but need minor revision:
-Add the statements of novelty at the end of introduction.
-Please, Add 2 figure mechanism of the topic that the manuscript discuss for.
We appreciate the reviewer’s insightful suggestion. We now added a statement of novelty at the end of the introduction. We have also included a figure that illustrates key mechanisms related to the impact of dietary patterns on breast cancer risk and prevention.
This manuscript is a resubmission of an earlier submission. The following is a list of the peer review reports and author responses from that submission.
Round 1
Reviewer 1 Report
Comments and Suggestions for Authors
Dear Authors.
Thank you for the opportunity to review your work. The role of nutrition as a modifiable risk factor is key in the prevention of breast cancer, the most frequently diagnosed tumour in women. After reviewing and analysing your work, I would like to make a couple of recommendations and suggestions. I would be grateful if you would take my comments into account as I believe that they can complete your manuscript.
- The summary talks about the search in different databases and the inclusion criteria taken into account (specifically only one, which is that they are in English). Were no other inclusion/exclusion criteria considered in the search? What search terms were considered to carry out the review?
- In the introduction, between lines 46-50, reference is made to different research. However, there is no bibliography reference.
- In lines 116 and 122, the reference should be placed after the "et al. (X)" and not at the end of the paragraph.
- On line 149 the reference is in APA style "According to Socha and Sobiech (2022),", modify to the format of the rest of the manuscript.
- In section 3.2 which consists of several sub-sections, only reference 9 is mentioned and it is placed after the full stop and not before it. It should be completed with a larger bibliography to give consistency and justification to the text.
Also, it goes from reference 9 to reference 12 (without including references 10 and 11 in the manuscript).
- In sections 3.5 and 3.6 there are no bibliographical references.
Best regards
Reviewer 2 Report
Comments and Suggestions for Authors
There are thirteen researchers who contributed to the preparation of this scientific review. When the references used in the preparation of the review are examined, it is seen that thirteen researchers do not have a study on the relevant subject. A researcher who writes a scientific review must have at least 5 studies on this subject. In addition, at least 70% of the references used must be from the last 5 years. 25 references are not enough for a review on this subject. The study is not suitable for publication.
Reviewer 3 Report
Comments and Suggestions for Authors
The review presents evidence in support of a single narrative idea that the authors have. It does not discuss studies with conflicting evidence and even those studies presented are not interpreted for their findings. For e.g "Lai-Yan Wong et al. conducted a systematic review and meta-analysis of prospective studies to examine the association between dietary patterns and the risk of breast cancer. By analyzing data from multiple studies, the researchers aimed to identify consistent patterns or correlations between specific dietary habits and the likelihood of developing breast cancer. Their findings provided valuable insights into the role of diet in breast cancer risk and highlighted potential dietary factors that may influence disease outcomes (8)."
The study is appreciated but the outcome and implications are hardly even mentioned.
Studies that provide conflicting evidence and any possible reasons for such conflicts need to be discussed.
Cultural and geographical reasons for dietary habits will need to be presented in detail. Figures indicating food groups and breast cancer risk, geography or tradition wise may be informative.
Alcohol consumption and its increase among women from cultures where taboos existed on women drinking and its contribution to altered nutritional status is not discussed in detail.
Although fewer studies exist on nutrition alone as risk factors, most studies would include such factors within their framework. Citing data or summarizing findings from such studies which may sometimes be presented along with other non-dietary risks will add the much needed depth to the review.
Comments on the Quality of English LanguageQuality of English is fine. Very few language related errors are present.